# Voice spoofing detection using a neural networks assembly considering spectrograms and mel frequency cepstral coefficients

Carlos Alberto Hernández-Nava[1], Eric Alfredo Rincón-García[2], Pedro Lara-Velázquez[2], Sergio Gerardo de-los-Cobos-Silva[2], Miguel Angel Gutiérrez-Andrade[2] and Roman Anselmo Mora-Gutiérrez[3]

[1] Posgrado en Ciencias y Tecnologías de la Información, Universidad Autónoma Metropolitana, Ciudad de México, Ciudad de México, México
[2] Departamento de Ingeniería Eléctrica, Universidad Autónoma Metropolitana, Ciudad de México, Ciudad de México, México
[3] Departamento de Sistemas, Universidad Autónoma Metropolitana de Azcapotzalco, Ciudad de México, Ciudad de México, México



## ABSTRACT

Nowadays, biometric authentication has gained relevance due to the technological advances that have allowed its inclusion in many daily-use devices. However, this same advantage has also brought dangers, as spoofing attacks are now more common. This work addresses the vulnerabilities of automatic speaker verification authentication systems, which are prone to attacks arising from new techniques for the generation of spoofed audio. In this article, we present a countermeasure for these attacks using an approach that includes easy to implement feature extractors such as spectrograms and mel frequency cepstral coefficients, as well as a modular architecture based on deep neural networks. Finally, we evaluate our proposal using the well-know ASVspoof 2017 V2 database, the experiments show that using the final architecture the best performance is obtained, achieving an equal error rate of 6.66% on the evaluation set.

## INTRODUCTION

The great growth of social networks in recent years is primarily attributed to the widespread accessibility to many different devices that facilitate the exchange of biometric information, such as computers, cell phones and tablets. These devices enable the transmission of images of human faces, full body videos, as well as audio recordings. Such information is used to train different tools capable of generating high-quality audiovisual media, mainly for entertainment purposes; however, due to the vast amount of information and the current power of these techniques, it is very difficult to distinguish between generated and genuine content. This generated material has found beneficial applications in a wide range of fields, including entertainment and, more recently, the

Corresponding author
Carlos Alberto Hernández-Nava, cahn@xanum.uam.mx

generation of diverse digital media on social networks; unfortunately, as mentioned in *Wu et al. (2015)*, it is also being exploited for fraudulent activities.

Today, technology has reached a level of maturity that enables biometric authentication across many different applications and devices. However, it is essential to make an effort to safeguard against identity fraud attempts, especially considering the increasing number of generated media as mentioned in *Echizen et al. (2021)*. Specifically, automatic speaker verification (ASV) systems, which are frequently used for speaker verification in telephony, are prone to malicious authentication attempts since they rely solely on the received sound as the means of authentication.

Several models based on neural networks have been developed for the detection of generated or manipulated audios intended for identity theft. Initially, basic neural networks were employed for this purpose. However, as technology progressed, more sophisticated architectures were gradually adopted to enhance their performance in fulfilling this task.

Nowadays, the detection of media created with the intention of being used for counterfeiting has garnered significant attention from the community. As a result, applications are being developed to detect these types of counterfeit files. This work specifically focuses on the growing interest in developing countermeasures against identity theft through automatic speaker verification. The remaining sections of this work are organized as follows: the next subsection presents the most important works related to spoof detection; "Materials and Methods" describes our proposed methodology for spoof detection; "Results" outlines the experiments conducted; "Discussion" comprises the discussion derived from the findings and "Conclusion" presents the conclusions.

## Related work

Interest in biometric recognition of speech and speakers is not new. In 2007, a liveness verification system, based on lip movement, was proposed in *Faraj & Bigun (2007)* as a means of protection against identity theft attempts, particularly through the generation of videos for this purpose.

Given the importance of the problem, the establishment of a standardized database became necessary. In 2015, the National Institute of Informatics initiated two challenges, namely the Voice Conversion Challenge and the ASVspoof Challenge, with the aim of providing an evaluation platform and metrics to facilitate a fair comparison among the proposed techniques related to media cloning and detection.

The Voice Conversion Challenge (*Toda et al., 2016*) is a biennial event that started in 2016, in this challenge participants are provided with a database and tasked with developing voice converters using their own methods. The organizers then evaluate and classify the transformed speech submitted by the participants.

The ASVspoof challenge (*Wu et al., 2015*), which is also a biennial event, is highly relevant to this work. The challenge provides a database comprising pairs of genuine and false or generated audios, which participant's models must accurately classify. Since the release of the ASVspoof challenge databases, investigations have yielded remarkably favorable results. The latest version of the databases was published in 2021.

In *Zhang, Yu & Hansen (2017)*, the authors propose an architecture that combines convolutional neural networks (CNN) and recurrent neural networks (RNN) simultaneously. To evaluate the effectiveness of their method, they utilized the ASVspoof 2015 database, where the input of their model consisted of spectrograms extracted from the audios. Due to the widely varying durations in this database, the authors decided to standardize the duration to four seconds for all audio samples. Through their experiments, they achieved an equal error rate (EER) of 1.47%.

It is worth highlighting the efforts made to create new detection models. For instance, in *Lavrentyeva et al. (2017)*, the authors proposed a simplified version of the Light CNN architecture that employs Max Feature Map (MFM) activation. The authors implemented this network to classify audios into two possible outcomes: genuine or false, specifically to prevent spoofing, they reported an equal error rate of 6.73% in the ASVspoof 2017 database.

In addition to proposing novel models, another factor to consider is the level of complexity required for the proposals. In *Pang & He (2017)*, the authors demonstrated that the implementing very deep or complicated neural networks is not necessarily essential for impersonation detection in identity verification. They explained that satisfactory results can be achieved with simple models. Their model consisted of an input layer, two CNN layers, a gated recurrent unit (GRU) layer, and a final layer. Despite its apparent simplicity, this model yielded excellent performance, with an EER of 0.77% on a *corpus* of 28,000 audios extracted from the APSRD (Authentic and Playback Speaker Recognition Database).

The alternative approach involves employing increasingly deeper neural networks; however, this can give rise to the vanishing gradient problem. To address this difficulty, residual neural networks (ResNet) have emerged. They have proven to be successful in image recognition, as demonstrated in *Chen et al. (2017)*, where their effectiveness for spoofed audio detection was investigated. The evaluation of this research was conducted using the ASVspoof 2017 dataset, revealing that Resnet achieved one of the best performances among the systems employing a single model.

So far, it has not been mentioned whether high-quality audio is truly essential to carrying out an audio attack. However, *Lorenzo-Trueba et al. (2018)* explored the potential of utilizing solely low-quality data to train models against spoofing. For this purpose, they developed a system based on generative adversarial networks (GAN), to enhance the quality of audio files accessible on the internet.

Understanding the significance of audio quality is crucial to comprehend the nature of the challenges at hand. This is particularly relevant because voice-controlled devices (VCDs) including popular examples like Alexa, Siri, and others are increasingly prevalent. These devices are primarily utilized for automating home appliances and other entertainment tools. Since voice attacks do not necessitate high-quality audio, it can be inferred that these devices might be vulnerable to such attacks.

In an analysis conducted by *Malik et al. (2020)*, multi-hop replay attacks were shown to be a vulnerability since they are carried out using VCDs with the intention of accessing other VCDs connected to the internet. For example, a device could be used to replicate the

voice of the speaker, giving an order or command to a second VCD, and the latter would fulfill its function without verifying whether the instruction truly originates from the speaker or is simply a repetition of the voice of the user.

After establishing the vulnerability of VCDs, (*Gong, Yang & Poellabauer, 2020*) presents another concern regarding the number of channels employed in attacks on these devices. They present a neural network-based model designed for the specific purpose of detecting multichannel audio. The proposed model allows for an arbitrary number of input channels, in addition, what can be highlighted is that their model is fully developed in a neural network framework, enabling potential integration with other neural network-based models in the future.

Combinations of neural networks are frequently employed in recent and advanced studies. In *Dua, Jain & Kumar (2021)*, the authors examined the performance of various architectures, incorporating deep neural networks (DNNs), long short-term memory (LSTM) layers, temporal convolution (TC), and spatial convolution (SC). To evaluate the performance of their proposal, they used ASVspoof 2015 and ASVspoof 2019 databases, achieving good results in both, with particularly impressive results in the latter.

It is worth mentioning that there are two main research approaches for spoof detection. The first approach involves using diverse architectures and classification models, including the Gaussian mixture model (GMM), support-vector machines (SVM), neural networks such as CNN, RNN, LSTM, GAN, ResNet, and even autoencoders.

The second research approach involves using diverse techniques for audio feature extraction, including different types of spectrograms or coefficients such as mel frequency cepstral coefficients (MFCC), inverse mel-frequency cepstral coefficients (IMFCC), complex cepstral coefficients (CCC), linear frequency cepstral coefficients (LFC), constant Q cepstral coefficients (CQCC), teager energy cepstral coefficients (TECC), linear predictive cepstral coefficients (LPCC), as well as various combinations thereof.

Although some of the extraction techniques and proposed architectures are highly efficient, they can be complex to comprehend and replicate. Therefore, we believe it is crucial to propose a technique that is both easily understandable and reproducible, while maintaining high precision in the specific task of audio spoof detection.

## MATERIALS AND METHODS

To accurately classify genuine and spoofed audio, it is necessary to identify and extract useful information from them. In this study, we found that combining the use of spectrograms and mel-frequency cepstral coefficients is sufficient to achieve higher accuracy that the state-of-the-art methods.

### Spectrograms

To obtain the spectrograms, samples are taken through a time window to calculate the frequency content of the samples using the short time Fourier transform (STFT). This process involves extracting and analyzing several signal frames at each window

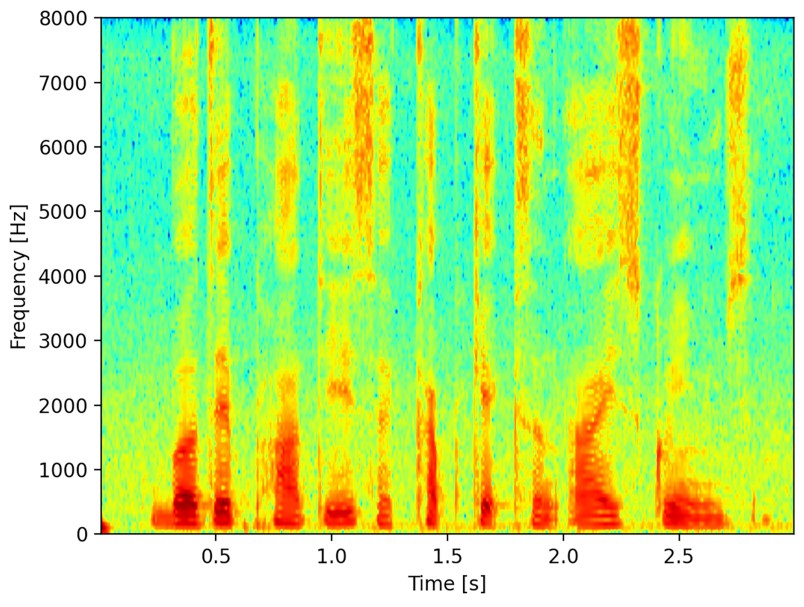

**Figure 1 Spectrogram.**

displacement over time. Each frame is added to a matrix that represents the variation in the spectrum and energy of the signal. As new frames are obtained, they are consecutively added to the first position in the array. In this way, the variation of the signal's spectrum and energy can be represented as a function of time.

After conducting tests, it became clear that linear spectrograms alone did not capture all the necessary information to distinguish between a spoof audio and a genuine one. Therefore, we decided to use spectrograms with a logarithmic scale to better capture audio information. As expected, after performing additional experiments, we found that both linear and logarithmic spectrograms were necessary to achieve high accuracy. In the following paragraphs, we provide more details on these findings.

Although there are many representations available, for this work, we considered the time on the abscissa axis as consecutive sequences of Fourier transforms, with the frequency expressed in Hz on the ordinate axis, and finally the representation of the energy expressed in dB represented with a color palette, Fig. 1 illustrates a linear spectrogram with time on the horizontal axis and frequency on the vertical axis.

The main modification for logarithmic spectrograms involves changing the scale of the ordinate axis from a linear to a logarithmic scale. For this study, we considered two different areas of interest. The first area includes values between $10^2$ and $10^4$, as shown in Fig. 2A. This interval was selected because it represents the range where the greatest amount of sound wave energy is concentrated for human voice. The second zone was reduced to values between $10^3$ and $10^4$ to focus on the zone that corresponds to the highest frequencies and exclude the lowest ones, as shown in Fig. 2B.

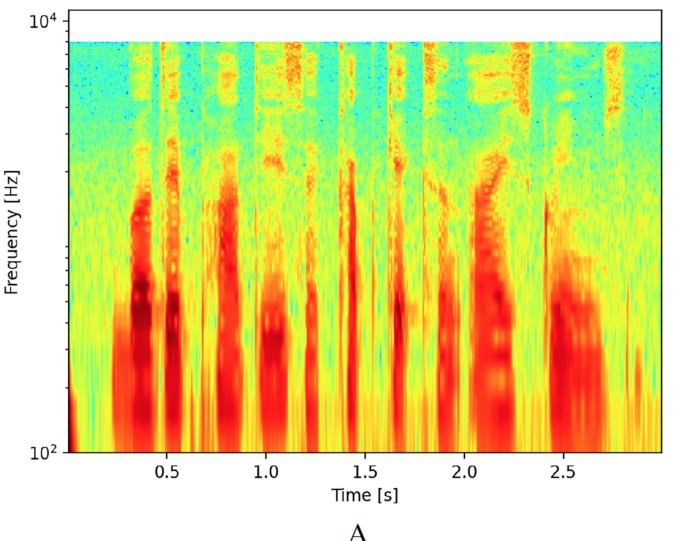
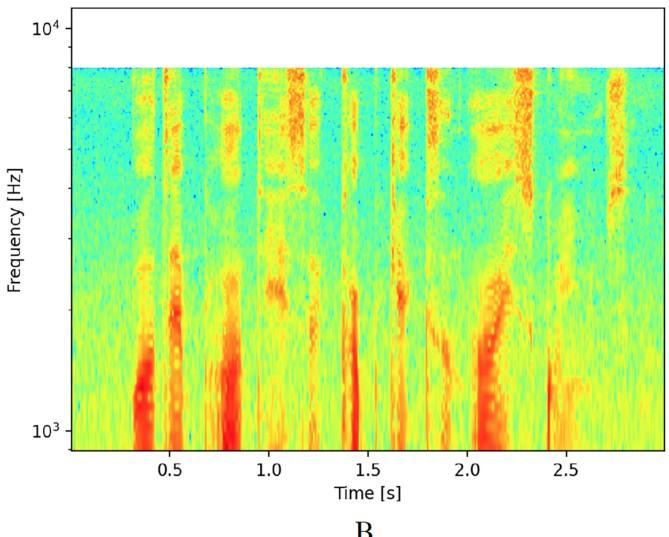

A
B

**Figure 2  Spectrograms with logarithmic scale.**

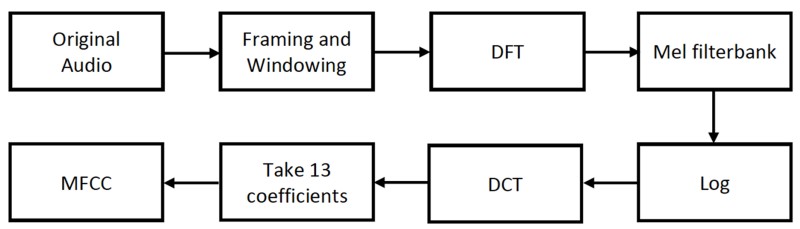

**Figure 3  MFCC feature extraction process.**

## MFCC

The most commonly reported feature extraction technique for spoof detection in the specialized literature is the mel frequency cepstral coefficients, originally proposed by *Davis & Mermelstein (1980)*. These are the widely used features to represent the human voice and have shown good results in various environments.

Prior to the extraction of MFCCs, an analog signal is converted into a digital signal through sampling at a specific sample rate. The digital signal is subjected to a series of processes, as illustrated in Fig. 3, to extract the MFCC features.

After dividing the analog signal into overlapping frames, the Discrete Fourier Transform (DFT) of the signal is calculated. Subsequently, the signal is filtered using the Mel filter bank, and the output is log compressed. It is then transformed into the cepstral domain using the Discrete Cosine Transform (DCT), preserving the first 13 coefficients while discarding the higher ones. As mentioned in *Kuamr, Dua & Choudhary (2014)*, this is because 13 coefficients are sufficient for representing the speech signal.

In this work, MFCCs are used in two different ways. First, 13 coefficients are extracted for each window, resulting in a $13 \times 298$ matrix. Subsequently, the matrix is reduced to a
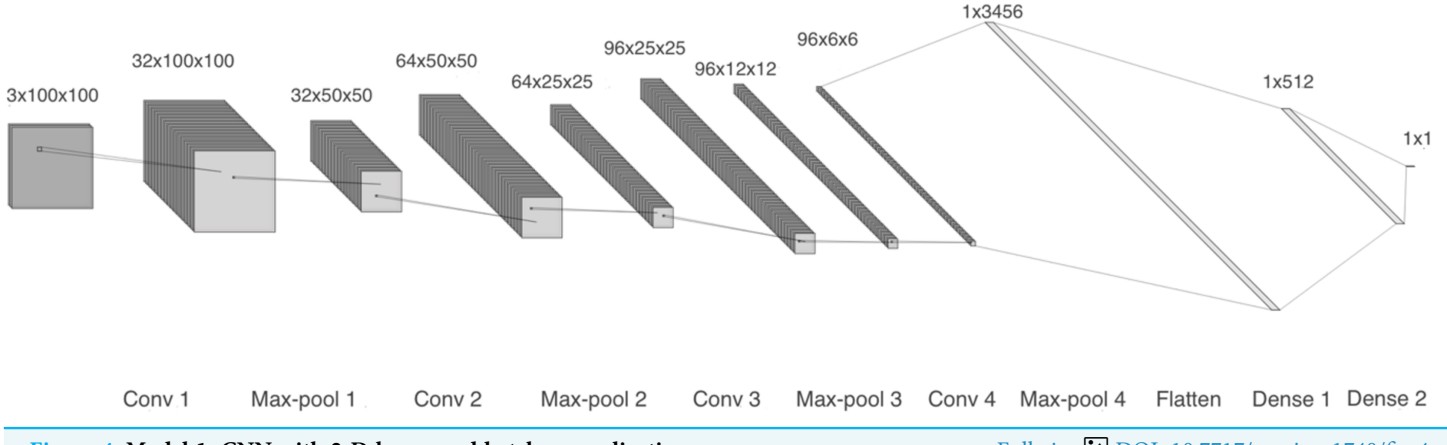

**Figure 4 Model 1, CNN with 2-D layers and batch normalization.**

vector of 13 coordinates, by calculating the average of all the windows. Both the matrix and the vector are used for audio classification.

## Models for spectrograms

To extract information from the spectrograms and properly classify the audios, we propose using two models based in convolutional neural networks.

The first model comprises four 2-D convolutional layers with 32, 64, 96, and 96 filters, respectively. After each convolution, batch normalization and max-pooling are performed. Finally, a flatten layer and two dense layers are included, as shown in Fig. 4. We trained two copies of model 1: one with linear spectrograms and the other with logarithmic spectrograms with values between $10^2$ and $10^4$.

For the linear spectrogram version, we used five epochs, a batch size of 200, a binary cross-entropy loss function, and RMSprop optimizer. In contrast, the logarithmic spectrogram version required 10 epochs, a smaller batch size of 20, the same loss function, and the Adam optimizer.

The second model was trained using logarithmic spectrograms with values between $10^3$ and $10^4$. The network comprises three time-distributed 2-D convolutional layers with 32, 64, and 96 filters respectively. After each convolution, time-distributed max-pooling is performed. Next, a flatten layer, a dense layer and a dropout layer are included. Finally, a dense layer completes the model, as shown in Fig. 5, 10 epochs were required to achieve good convergence with a batch size of 10, we used a binary cross-entropy loss function and the Adam optimizer.

## Models for MFCCs

For the MFCCs coefficients, we used two additional models.

Model 3 takes the vector representation of the MFCCs as input and passes the 13 coefficients through three time distributed 1-D dense layers with 24, 13 and 10 units, respectively, along with a dropout layer. Next, we added three LSTM layers with 10, 15 and
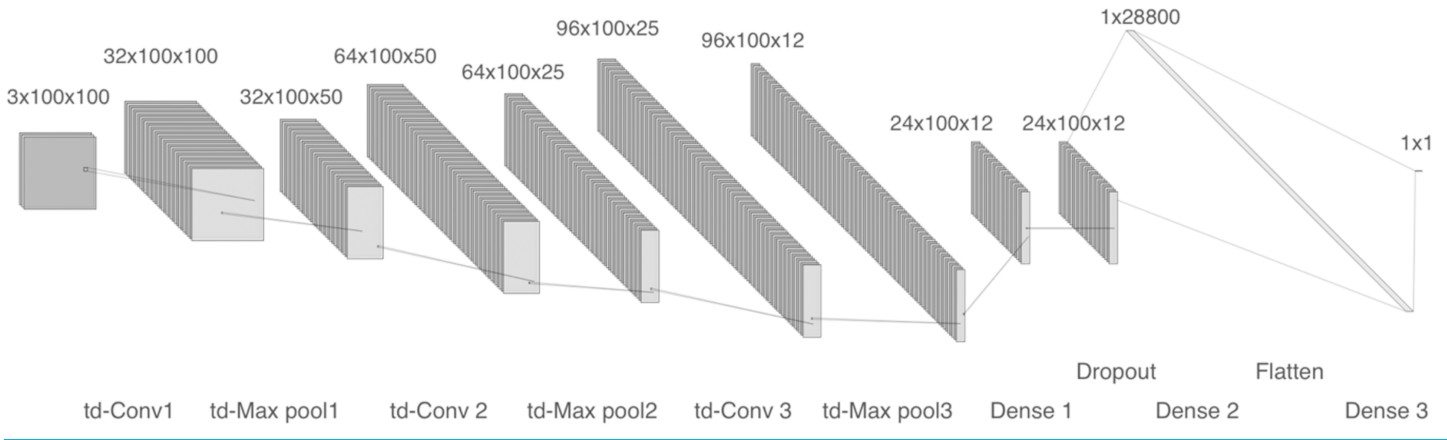

**Figure 5 Model 2, CNN with tD 2-D layers and dropout.**

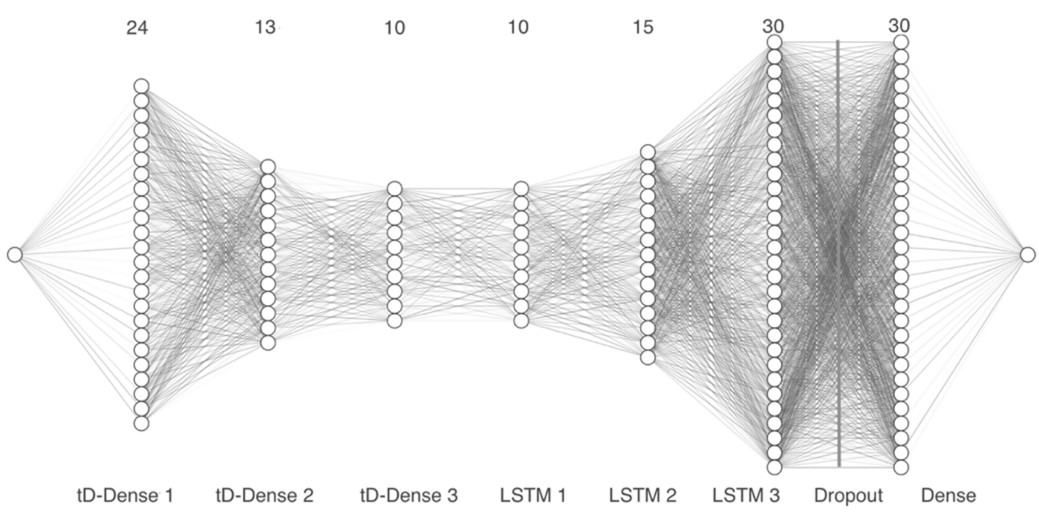

**Figure 6 Model 3, CNN with tD and convolutional 1-D layers.**

30 units, followed by a second dropout layer. Finally, a dense layer with 30 units was included, as can be seen in Fig. 6.

Model 4 receives the two-dimensional MFCCs as input. The network comprises a time distributed 2-D convolutional layer with 36 filters, followed by a batch normalization layer, a second 2-D convolutional layer with 64 filters, and a max pooling layer. We then included two dense layers with 24 units each, followed by a flatten layer and a dense layer, as can be seen in Fig. 7.

For the models using MFCCs, more epochs could be used due to the reduced feature extraction size. For the model that uses a vector, we employed 60 epochs, a batch size of 200, the binary cross-entropy loss function, and the Adam optimizer. As for the version using MFCCs in matrix form, it required 30 epochs for better convergence, a large batch size of 300, the same loss function, and the RMSprop optimizer, it should be noted that for

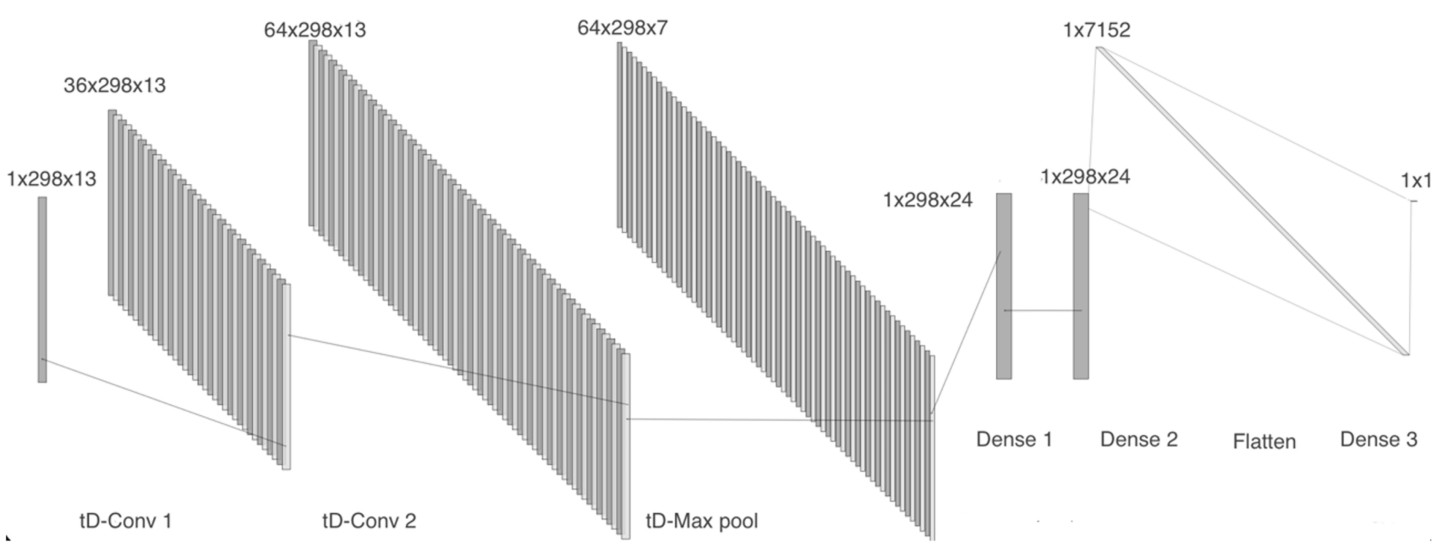

**Figure 7  Model 4, CNN with tD 2-D layers to work with MFCCs.**

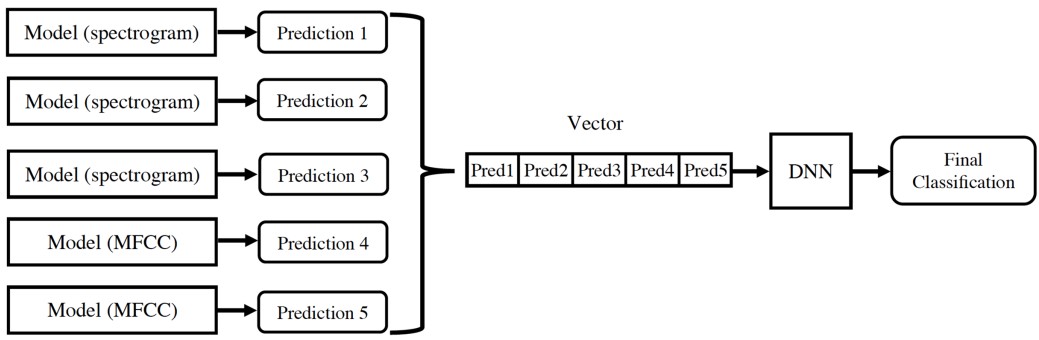

**Figure 8  Design of information flow in the final architecture.**

all these models, whether using spectrograms or MFCCs, a learning rate of 0.001 was applied.

For each audio, the linear spectrogram, the two logarithmic spectrograms described above, and the MFCC coefficients are calculated and introduced into their respective models. The classification results from each model are concatenated in a vector, which is used as input for a new neural network. This provides the final classification prediction, as depicted in Fig. 8.

All models were individually trained and subsequently assembled to form the final architecture, as illustrated in Fig. 9. This design offers a significant advantage because it allows for the seamless addition of new models and feature extractions to the architecture. This flexibility arises from the way they are integrated to produce the final classification for each audio.

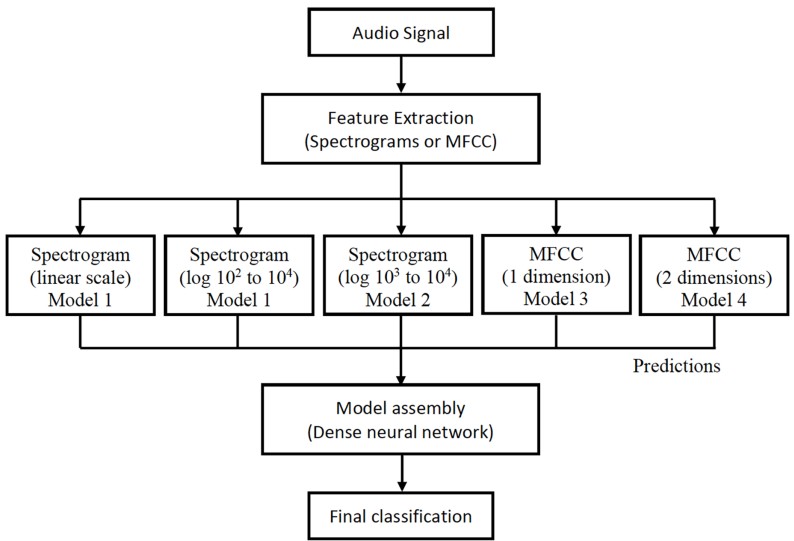

**Figure 9 Final architecture of the assembly of the proposed models.**

# RESULTS

To evaluate the performance of our proposed methodology, we utilized the ASVspoof 2017 V2 database, which is the second version of the database used in the ASVspoof 2017 challenge (*Kinnunen et al., 2017*; *Delgado et al., 2018*). This database is focused on replay attacks, which are generated by recording the voice of a genuine speaker and then replaying it to an ASV system, instead of using the genuine speech of the person.

The genuine audios are a subset of the RedDots4 *corpus* (*Lee et al., 2015*), collected by volunteers using Android smartphones. In contrast, the spoofed audios result from playing and recording authentic audios using a variety of devices and within different acoustic environments, that is, the physical space in which the recording process occurs.

The ASVspoof 2017 V2 database comprises recordings collected with various replay configurations, each involving a playback device, an acoustic environment, and a recording device. This database includes 61 different configurations, created by combining 26 environments, 26 playback devices, and 25 recording devices, these configurations were applied to the speech of 42 speakers, as detailed in Table 1.

The *corpus* consists of 18,030 audio files of varying durations, divided into three datasets, the first dataset contains 1,507 genuine and 1,507 generated audios for training. The second dataset consists of 760 genuine and 950 generated audios for development. Finally, the third dataset includes 1,298 genuine and 12,008 generated audio files for evaluation.

To homogenize the audios, we decided to make them all have a duration of 3 s. This decision is based on the findings of *Zhang, Yu & Hansen (2017)*, where it was mentioned that if the audios have a duration of less than 2.5 s, favorable results are not obtained. Furthermore, after analyzing the database, we found that most of the audios have a

**Table 1  Description of the ASVspoof 2017 V2 database.**

| Subset | Speakers | Replay sessions | Replay config | Genuine audio | Spoof audio |
|---|---|---|---|---|---|
| Training | 10 | 6 | 3 | 1,507 | 1,507 |
| Development | 8 | 10 | 10 | 760 | 950 |
| Evaluation | 24 | 161 | 57 | 1,298 | 12,008 |

**Table 2  Accuracy and EER for each model.**

| Feature extraction | Training accuracy (%) | Evaluation accuracy (%) | EER (%) |
|---|---|---|---|
| Linear spectrograms | 99.41 | 72.67 | 22.67 |
| Log spectrograms $(10^2, 10^4)$ | 93.20 | 64.02 | 23.13 |
| Log spectrograms $(10^3, 10^4)$ | 92.20 | 70.71 | 26.12 |
| MFCC (vector) | 98.11 | 68.98 | 21.06 |
| MFCC (matrix) | 87.93 | 81.42 | 22.80 |

**Table 3  Accuracy and EER for assembly.**

| Model | Evaluation accuracy (%) | EER (%) |
|---|---|---|
| Assembly | 96.46 | 6.66 |

duration close to 3 s, therefore, we determine that a duration of 3 s is sufficient to achieve a good performance.

In this work, when an audio was shorter, silence was added, and for longer audios only the initial 3 s were used. This strategy served as a preprocessing step for all the feature extraction techniques discussed in this document. A similar strategy is employed by *Pang & He (2017)*, where if the sample is too long, they simply cut the excess part, and if the sample is too short, they concatenate the original sample with itself to obtain a desired length.

The periods of silence or 0 values in some of the original RedDots database audio files could potentially impact the performance of detection systems. During the creation of playback attacks, these periods of silence were distorted into other values, making the task more difficult, this issue was mitigated with the release of the ASVspoof 2017 V2 database.

In this context, both truncation and addition of repeated audio or silence can indeed affect the results. *Kwak et al. (2021)* demonstrate in their experiments that a minimum of 3 s is necessary to achieve optimal system performance. However, for this specific proposal, we also needed to establish a fixed duration to ensure consistent spectrogram scales and maintain image homogeneity.

The performance of the proposed methodology was evaluated using the Equal Error Rate or Crossover Error Rate (CER), which represents the point at which the false rejection rate (FRR) and the false acceptance rate (FAR) are equal. It is expected that higher accuracy

**Table 4 Comparison of proposed approach with existing techniques, in the ASVspoof 2017 V2 database.**

| Approach | Feature extraction | Classifier | Eval EER (%) |
|---|---|---|---|
| **Proposed assembly** | Spectrograms + MFCC | DNN | **6.66** |
| *Wickramasinghe et al. (2019)* | CF + CM | GMM | 8.58 |
| *Das & Li (2018)* | CQCC + IFCC, DCTILPR + RMFCC | GMM | 9.01 |
| *Suthokumar et al. (2019)* | PPRFWS_LR | GMM | 9.28 |
| *Jelil et al. (2018)* | CQCC + CILPR | GMM | 9.41 |
| *Kamble & Patil (2021)* | CQCC + LFCC + MFCC + TECC | GMM | 10.45 |
| *Balamurali et al. (2019)* | MFCC + IMFCC + CQCC + CCC + RFCC + LFCC + LPCC + Spectrogram + Autoencoder Features | GMM | 10.80 |
| *Delgado et al. (2018)* | CQCC (baseline) | GMM | 12.24 |
| *Kamble, Tak & Patil (2020)* | CQCC + ESA-IFCC | GMM | 12.93 |
| *Tapkir et al. (2018)* | CQCC + PNCC | GMM | 12.98 |
| *Yang, Das & Li (2018)* | eCQCC-DA | DNN | 13.38 |

**Note:**
The result for the proposed assembly is shown in bold.

will result in a lower EER, as optimal performance is achieved with 100% accuracy or an EER equal to 0.

As previously mentioned, all models were individually trained and tested before being assembled into the final architecture. Table 2 presents the accuracy and EER (expressed as a percentage) achieved by each proposed model.

The next step involves assembling the five feature extraction processes and their corresponding models to establish a correspondence rule between the audios and their respective classification. As expected, the final architecture outperforms the individual models when they work independently, as evidenced by the results in Table 3, where an accuracy of 96.46% and an EER of 6.66% were achieved.

## DISCUSSION

To properly evaluate our proposal, we compared it with other models reported in literature that used the ASVspoof 2017 V2 database. Table 4 presents the EER reported by different authors, along with the extraction techniques and the classifiers used by them. Our proposal achieved the lowest EER among all the models compared. Thus, our methodology, which incorporates mel frequency cepstral coefficients and linear and logarithmic spectrograms, along with CNNs, exceeds state-of-the-art results.

Individually, the proposed models exhibit adequate but not outstanding behavior. Most of the models in this research extract an image directly from the audio files, specifically spectrograms. Although MFCCs are not images, they generate a matrix that can be treated as an image, enabling us to harness the capabilities of CNNs for tasks involving image processing, such as segmentation and recognition.

It is worth highlighting the importance of preprocessing the audio files to obtain a more homogeneous database to perform the feature extraction. This ensures that the features

serve their intended purpose in training the models, even when dealing with the unbalanced ASVspoof 2017 V2 database used in this work.

The quality of performance can be verified by examining the EER metric, where the final architecture achieves a value of 6.66%. This value allow us to gauge our performance in relation to other works and proposals. It can also be described as a classification accuracy of 96.46% out of the eighteen thousand audio samples.

## CONCLUSIONS

As artificial intelligence continues to improve, it becomes increasingly easier to generate spoofed audio that is more challenging to detect. In this article, we describe a methodology that can be easily implemented and achieves high accuracy in detecting spoofed audio. Our findings suggest that MFCCs and linear and logarithmic spectrograms are sufficient to achieve outstanding performance, and the advantage is that these features can be easily calculated using established libraries. Finally, the information is processed using either CNN or DNN, and the classification is completed with only a small number of misclassified audio files.

The strength of this proposal is achieved not only by combining various techniques for extracting useful information from audio, but also by proposing different neural network architectures to be used with each technique, which highlights the importance of using a tailored approach for each technique.

Indeed, the truly outstanding result is achieved by the architecture of the final assembly, which merges the predictions given by each proposed model and generates a final classification. This assembly has not only proven to have an adequate behavior but also is above the cutting-edge results.

### Funding
This work was supported by Autonomous Metropolitan University. The funders had no role in study design, data collection and analysis, decision to publish, or preparation of the manuscript.

### Grant Disclosures
The following grant information was disclosed by the authors:
Autonomous Metropolitan University.

### Competing Interests
The authors declare that they have no competing interests.

### Author Contributions
- Carlos Alberto Hernández-Nava conceived and designed the experiments, performed the experiments, performed the computation work, prepared figures and/or tables, authored or reviewed drafts of the article, and approved the final draft.

- Eric Alfredo Rincón-García conceived and designed the experiments, performed the computation work, authored or reviewed drafts of the article, and approved the final draft.
- Pedro Lara-Velázquez conceived and designed the experiments, performed the computation work, authored or reviewed drafts of the article, and approved the final draft.
- Sergio Gerardo de-los-Cobos-Silva analyzed the data, authored or reviewed drafts of the article, and approved the final draft.
- Miguel Angel Gutiérrez-Andrade analyzed the data, authored or reviewed drafts of the article, and approved the final draft.
- Roman Anselmo Mora-Gutiérrez analyzed the data, authored or reviewed drafts of the article, and approved the final draft.

## Data Availability

The database is available at The 2nd Automatic Speaker Verification Spoofing and Countermeasures Challenge (ASVspoof 2017) Database Version 2, https://doi.org/10.7488/ds/2332.

The code is available at: CAHNProtium/Spectrogram-and-mfcc: Assembly Final (Assembly). Zenodo. https://doi.org/10.5281/zenodo.8195332.

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
