# Peer review of "Voice spoofing detection using a neural networks assembly considering spectrograms and mel frequency cepstral coefficients"

_PeerJ Computer Science, doi:10.7717/peerj-cs.1740_

## Round 0.1 · original submission · Minor Revisions

The spell error and grammar may also be corrected.

**Language Note:** The Academic Editor has identified that the English language must be improved. PeerJ can provide language editing services - please contact us at copyediting@peerj.com for pricing (be sure to provide your manuscript number and title). Alternatively, you should make your own arrangements to improve the language quality and provide details in your response letter. – PeerJ Staff

·

Basic reporting

The literature review is comprehensive and provides a historical context for the research. It discusses various related works and their approaches to spoof detection, highlighting the evolution of techniques over time.
The Results section presents the performance metrics of the proposed models and the final architecture. It is well-organized and provides a clear picture of the methodology's effectiveness.
Overall, the paper effectively compares the proposed methodology's performance with existing models in the literature, highlighting the approach's strengths.

Experimental design

The article is highly relevant in biometric authentication, as spoofing attacks on speaker verification systems have become a significant concern. The authors do an excellent job of explaining the importance of their research in addressing these vulnerabilities.
The methodology is well-detailed and structured. It describes the feature extraction techniques (MFCCs and spectrograms) and the neural network architectures used in a clear and organized manner. The inclusion of figures and diagrams enhances the understanding of the proposed models.

Validity of the findings

1. The article discusses the challenges and countermeasures related to spoofing attacks in automatic speaker verification systems. While the problem is not entirely novel, the proposed methodology, combining Mel Frequency Cepstral Coefficients (MFCCs) and spectrograms with convolutional neural networks (CNNs), seems to be a unique approach.
2. Using the ASVspoof 2017 V2 database for evaluation adds credibility to the results.

Minor observation:
3. Clarifying the process of assembling the different models into the final architecture would benefit the article more. Explain how the outputs of the individual models are combined for classification, and providing a diagram or flowchart of this process that would enhance understanding.

Additional comments

Minor Suggestions:
These minor suggestions might benefit the article and its readers more but are not necessary to be incorporated as the paper is well-structured and contains ample information.

1. More information about the specific hyperparameters used in training the neural networks, such as learning or dropout rates, would enhance the reproducibility and scientific rigor of the work.
2. The article could benefit from a more explicit mention of the results achieved in terms of accuracy and Equal Error Rate (EER) to provide a more precise summary of the paper's contributions.

Reviewer 2 ·

Basic reporting

Overall, the text effectively explains the process of spectrogram generation and sets the stage for a more detailed discussion of the research findings regarding the use of linear and logarithmic spectrograms to distinguish between genuine and spoof audio.

Experimental design

I have few questions:
The use of the ASVspoof 2017 V2 database is reasonable, especially for addressing replay attacks. However, some questions may arise regarding the representativeness of this dataset. For instance, how diverse is the dataset in terms of speaker characteristics, recording environments, and types of replay attacks?
The description of the dataset composition (Table 1) is helpful in understanding the size of each dataset for training, development, and evaluation. However, it would be valuable to know how the genuine and generated audio samples were selected or created. Were they randomly sampled or generated in a specific manner?

Validity of the findings

The addition of silence for shorter audios and truncation for longer audios is a common preprocessing step, as mentioned. It's relevant to understand whether these modifications impact the data's integrity and what implications they have for the proposed methodology's performance.

Additional comments

The paper, as presented, provides a well-written foundation for research in the field of audio signal analysis, specifically focusing on distinguishing between genuine and spoof audio, with a clear methodology and data description. With the suggested clarifications and additional information as outlined in the previous review, the paper can be considered for publication

---

## Round 0.2 · accepted · Accept

The comments have been addressed.

·

Basic reporting

None

Experimental design

None

Validity of the findings

None

Additional comments

None

Reviewer 2 ·

Basic reporting

The authors have revised the manuscript in a clear and unambiguous order.

Experimental design

No comments

Validity of the findings

Now it is clear

Additional comments

None